# Research Progress on Neuroprotective Effects of Isoquinoline Alkaloids

**DOI:** 10.3390/molecules28124797

**Published:** 2023-06-16

**Authors:** Jinhua Li, Yarong Wu, Shuze Dong, Ye Yu, Yuhao Wu, Benhan Xiang, Qin Li

**Affiliations:** 1School of Pharmacy, Hangzhou Medical College, Hangzhou 310013, China; 881012022216@hmc.edu.cn (J.L.); wyr18509023382@163.com (Y.W.); shuzedong2023@163.com (S.D.); 13346186957@163.com (Y.Y.); 15558662365@163.com (Y.W.); xiangbh2100@163.com (B.X.); 2Key Laboratory of Neuropsychiatric Drug Research of Zhejiang Province, Hangzhou 310013, China

**Keywords:** isoquinoline alkaloids, neuroprotection, nerve injury, mechanism

## Abstract

Neuronal injury and apoptosis are important causes of the occurrence and development of many neurodegenerative diseases, such as cerebral ischemia, Alzheimer’s disease, and Parkinson’s disease. Although the detailed mechanism of some diseases is unknown, the loss of neurons in the brain is still the main pathological feature. By exerting the neuroprotective effects of drugs, it is of great significance to alleviate the symptoms and improve the prognosis of these diseases. Isoquinoline alkaloids are important active ingredients in many traditional Chinese medicines. These substances have a wide range of pharmacological effects and significant activity. Although some studies have suggested that isoquinoline alkaloids may have pharmacological activities for treating neurodegenerative diseases, there is currently a lack of a comprehensive summary regarding their mechanisms and characteristics in neuroprotection. This paper provides a comprehensive review of the active components found in isoquinoline alkaloids that have neuroprotective effects. It thoroughly explains the various mechanisms behind the neuroprotective effects of isoquinoline alkaloids and summarizes their common characteristics. This information can serve as a reference for further research on the neuroprotective effects of isoquinoline alkaloids.

## 1. Introduction

Degenerative diseases of the nervous system have brought a significant medical and public health burden on people worldwide. With the acceleration of population aging, the prevalence and incidence of degenerative nervous system diseases have sharply increased in recent years [1]. Although the pathological mechanisms of cerebral ischemia, Alzheimer’s disease, and Parkinson’s disease differ, nerve cell injury and apoptosis are common pathological features [2]. After experiencing cerebral ischemia, the blood vessels and tissues in the brain may become damaged, leading to the destruction of the blood–brain barrier. Under the combined action of a series of cascade reactions, such as oxidative stress stimulation, the release of inflammatory factors, aggravation of autophagy, and mitochondrial dysfunction, a large number of nerve cells rapidly undergo apoptosis. This process directly leads to learning and cognitive dysfunction after cerebral ischemia [3]. The pathological mechanism of Alzheimer’s disease is still controversial, but studies have confirmed that progressive neurodegeneration is its main feature. Neuroinflammation and angiogenesis, neurogenesis, and neurological recovery dysfunction play an important role in the pathophysiology of AD [4]. The main cause of Parkinson’s disease is the degeneration and eventual loss of dopaminergic neurons in the substantia nigra. Dopamine replacement therapy has achieved remarkable results [5]. Therefore, drug research with neuroprotective effects, which can reduce nerve damage and promote neuronal repair and regeneration, is of great significance in alleviating the progression of such diseases and even curing them. However, the research progress of drugs with neuroprotective effects has been slow, and few such drugs have been marketed in recent years [6].

Isoquinoline alkaloids, including nuciferine, berberine, and tetrandrine, are the main active components extracted from traditional Chinese medicines, such as lotus leaf, Coptis chinensis Franch, and Stephania tetrandra S. Moore, respectively. Most of these traditional Chinese medicines have been used in clinical treatments for many years. The core functions are clearing heat and detoxifying, diuresis and detumescence, and promoting blood circulation and cooling the blood. Now, they have also been studied for their anti-inflammatory and anti-oxidant properties [7,8,9,10]. Upon observing their structures, we have found that they have a common structural basis and belong to isoquinoline alkaloids. By consulting the relevant literature on these alkaloids, it can be found that these compounds have a wide range of pharmacological effects and significant pharmacological activities, especially in improving nerve damage caused by neurodegenerative diseases. Studies have shown that the active ingredients of isoquinoline alkaloids can exert neuroprotective effects by inhibiting nerve injury inflammation, anti-oxidative damage, regulating autophagy, inhibiting intracellular calcium overload, and improving mitochondrial dysfunction, as well as promoting vascular endothelial cell proliferation, and neuronal repair and regeneration [11]. However, these alkaloids have not yet been used to treat neurodegenerative diseases, and the related research is not deep enough. Additionally, there is no review focused on a comprehensive summary of the neuroprotective effects of isoquinoline alkaloids.

Therefore, we used isoquinoline alkaloids, neuroprotection, nerve injury, and the names of some isoquinoline alkaloids, such as nuciferine, berberine, and tetrandrine, as well as some nerve injury mechanisms such as neuroinflammation, oxidative stress, and autophagy as keywords. After consulting a large amount of literature in PubMed and CNKI literature retrieval media, we have compiled a list of several representative isoquinoline alkaloids with neuroprotective effects, and comprehensively discuss their different mechanisms of neuroprotective effects and some common characteristics. It is hoped that this can provide a reference for further basic research on the neuroprotective effects of isoquinoline alkaloids.

## 2. Isoquinoline Alkaloids with Neuroprotective Effects

### 2.1. Nuciferine

Nuciferine is a natural product extracted from the traditional Chinese medicine, lotus leaf. Lotus leaf has the effect of dispelling dampness and decreasing phlegm, dispersing stasis, and lowering fever in the theory of traditional Chinese medicine. It is a traditional medicinal material for food therapy [12]. Nuciferine has been shown to have anti-inflammatory, antioxidant, anti-aging, hypolipidemic, and other pharmacological activities [13]. The physicochemical properties and metabolomics analysis of nuciferine showed that its absolute bioavailability is 69.56%. It has good fat solubility and high blood–brain barrier permeability, which are the bases for its treatment of central nervous system diseases [14]. At the same time, existing studies have also shown that nuciferine can produce neuroprotective effects by inhibiting neuroinflammation, reducing oxidative damage, and regulating autophagy [15].

### 2.2. Berberine

Berberine is a plant alkaloid extracted from Coptis chinensis Franch, which has a variety of pharmacological properties. Many studies have focused on lowering blood glucose, anti-obesity, and improving insulin resistance [16,17,18]. However, current studies have also shown that it has many pharmacological activities, such as anti-inflammatory, anticancer, cardiovascular, and central nervous system effects [19]. It is also a potential drug for the multitarget treatment of neurodegenerative diseases caused by metabolic disorders [20]. Usually, when neuronal damage is caused by neurometabolic diseases, multiple important organelles, including lysosomes, peroxisomes, and mitochondria, will undergo dysfunction, causing metabolic disorders and lead to a significant amount of neuronal apoptosis [21]. This feature of berberine provides a novel therapeutic idea for neuroprotection by improving metabolic disorders and optimizing energy metabolism.

### 2.3. Tetrandrine

Radix Stephanie Tetrandrine has been traditionally used as a medicine for dispelling wind and dampness, promoting diuresis, and reducing swelling. As the main alkaloid component of Radix Stephanie tetrandrae, tetrandrine has also received extensive attention in recent years [22]. Studies have demonstrated that this compound has a variety of neuroprotective pharmacological activities. The main neuroprotective mechanism involves regulating Ca^2+^ and K^+^ channels, maintaining intracellular calcium homeostasis, and reducing neuronal and glial cell damage caused by Ca^2+^ overload [23]. In addition, it can regulate central neurotransmitter transport and metabolism, inhibit neuroinflammation, improve vascular endothelial dysfunction, decrease oxidative stress, and regulate autophagy [24,25,26]. It has a highly comprehensive neuroprotective effect and is a potential neuroprotective agent.

### 2.4. Morphine

Morphine is a classic analgesic and sedative drug. At present, its clinical application is mainly to alleviate pain. Traditional studies have shown that morphine has a potent inhibitory effect on the central nervous system, and large doses of morphine can produce toxic effects on nerve cells [27]. However, we are cognizant that in recent years, many studies have shown that morphine can also exert neuroprotective effects through various mechanisms [28]. The main mechanisms include reducing intracellular Ca^2+^ overload, reducing cell oxidative damage, activating autophagy, and inhibiting intracellular toxic protein production to promote neuronal regeneration and differentiation [29,30,31,32]. At the same time, researchers have applied very low doses of morphine to very preterm infants, which has effectively improved the neurodevelopmental status of the infants [33]. This also reflects the neuroprotective potential of low-dose morphine. What’s more, studies have shown that morphine derivatives also have neuroprotective effects. For example, Hydrophone protects hippocampal CA1 neurons from ischemia-reperfusion injury by activating the mTOR signaling pathway, and methadone can improve cognitive dysfunction in drug withdrawal patients [34,35]. This suggests that derivatives of isoquinoline alkaloids may also have neuroprotective effects, but research in this area is more limited, and more related research is needed.

### 2.5. Tetrahydropalmatine

Tetrahydropalmatine is a key component of the traditional Chinese medicine Corydalis ambigua. In addition to the traditional analgesic, sedative, and hypnotic effects, many studies on the effects on the cardiovascular and central nervous systems have also made some progress [36,37]. Especially for neuroprotection, tetrahydropalmatine can not only inhibit the level of nerve injury inflammation, antioxidant damage, and regulate autophagy to reduce neuronal apoptosis, but also has a unique ability to promote vascular endothelial cell proliferation and neuronal repair and regeneration [38]. Angiogenesis plays an important role in wound healing, nerve defects, and nerve regeneration. Tetrahydropalmatine also has the potential to increase the expression of neurotrophic factors, promoting neurogenesis and enhancing the repair and regeneration abilities of neurons after injury [39].

The structures of several aforementioned alkaloids are shown in Figure 1. These structures all contain the same isoquinoline nucleus, which belongs to the active components of isoquinoline alkaloids. In addition to the above, a variety of isoquinoline alkaloids have neuroprotective effects, showing pharmacological activity in the treatment of neurodegenerative diseases [40]. In order to better demonstrate the neuroprotective effects and mechanisms of this type of compound, we selected about 10 representative isoquinoline alkaloids with the same isoquinoline nucleus structure and neuroprotective effect, as shown in Table 1.

## 3. Neuroprotective Effect and Mechanism of Isoquinoline Alkaloids

The above studies have shown that a variety of isoquinoline alkaloids with a common structural basis, can exert neuroprotective effects based on a variety of different modes of action and have potential application value for the treatment of many neurodegenerative diseases. Recent reports have also revealed the intracellular signaling pathways and mechanisms that these alkaloids may target, thereby exerting their neuroprotective effects. In this section, we will briefly describe the mechanism of nerve injury, summarize the neuroprotective mechanisms and signaling pathways of the five common isoquinoline alkaloids mentioned in the previous section, and clarify their potential contribution to the treatment of neurodegenerative diseases.

### 3.1. Neuroprotection towards Inflammatory Injury

Almost all neurodegenerative diseases are accompanied by inflammation. When neurons are affected under the action of various intracellular mechanisms, abnormal protein metabolism and degeneration, organelle dysfunction, etc., will occur, resulting in a large amount of neuronal apoptosis [91]. Inflammation is usually regarded as a process of self-repair of the body, which helps to remove a large number of fragments caused by cell necrosis and death. Therefore, neuronal apoptosis will induce the activation of immune cells and the release of a large number of inflammatory factors. However, due to inflammation, brain tissue damage, blood–brain barrier damage, and edema, cerebrovascular dysfunction will worsen and induce further neuronal apoptosis [92]. Therefore, although inflammation is not the main pathological mechanism of most neurodegenerative diseases, it is a key event in these diseases. Reducing inflammatory damage is still of great significance for neuroprotection and disease treatment [93]. As shown in Figure 2, isoquinoline alkaloids have significant anti-inflammatory activity. It has been shown that tetrandrine can effectively inhibit the activation of NLRP3 inflammasome by inhibiting I/R-induced overexpression of inflammatory and apoptotic factors such as NLRP3, caspase-1, IL-1β, IL-18, and Sirt-1. Up-regulation of Sirt-3 further inhibited NLRP1 inflammasome activation and significantly reduced the neurological deficit, infarct volume, and brain water content in MCAO mice [94]. In addition, it can also down-regulate the expression of NSE, TNF-α, NF-κB, TRAF1, GADD34, p-PERK, IRE1α, CHOP, and p-JNK by regulating the IRE1α/JNK/CHOP signaling pathway, reducing the level of neuroinflammation and neuronal apoptosis in mice with traumatic brain injury (TBI) [95]. Studies have shown that tetrandrine can reverse the ectopic transcription of inflammation-related genes, including TNFα, IL-1β, IL-6, COX-2, iNOS, and p65, in 5XFAD mice (a transgenic model of AD), inhibiting the secretion of inflammatory cytokines TLR4, p65, iNOS, and COX-2 in microglia BV2 cells induced by Aβ1-42, and improving AD by reducing inflammation and neurotoxicity [96].

In addition to tetrandrine, berberine and nuciferine also have good anti-inflammatory activity. In addition to activating the PI3K-AKT signaling pathway and inhibiting the NF-kB signaling pathway to produce anti-inflammatory effects, berberine can also inhibit the production of TNF-α, cyclooxygenase-2 (COX-2), and inducible nitric oxide synthase (iNOS), which helps to reduce neuroinflammation and prevent blood–brain barrier damage [97,98,99,100]. In addition to improving neuroinflammatory damage, berberine can also directly act on improving cognitive dysfunction in AD through a variety of other mechanisms. Nuciferine has been shown to inhibit the activation of NF-κB, reduce the release of pro-inflammatory mediators such as TNF-α, IL-1β, and PGE2, and inhibit the inflammatory response of BV2 cells induced by LPS. Further studies have shown that nuciferine can also activate PPAR-γ, inhibit neuroinflammatory damage caused by NF-kB, and exert neuroprotective effects [101].

### 3.2. Neuroprotection towards Oxidative Stress

The relationship between free radical damage and inflammatory damage is inseparable, and there is a complex interaction between them [102]. Cells produce free radicals during normal metabolism, and the body contains antioxidants that neutralize these free radicals, sustaining a dynamic balance between antioxidants and free radicals. When pathological damage causes an excessive accumulation of free radicals, the balance is disrupted, and oxidative stress occurs. Inflammatory damage is one of the factors that can trigger oxidative stress [103]. However, oxidative stress can cause cell and tissue damage, which can activate a variety of transcription factors, leading to abnormal expression of genes involved in inflammatory pathways and promoting inflammatory responses. The cycle of the two further exacerbates the disease [104]. As shown in Figure 3, in addition to anti-inflammatory activity, isoquinoline alkaloids have significant antioxidant activity, and berberine can protect C17.2 neural stem cells from oxidative damage [105]. The main mechanism involves reducing the level of reactive oxygen species (ROS) in C17.2 cells through the (NRf1/2)-(NQO-1)-(HO-1) pathway while down-regulating the apoptosis factors caspase-3 and Bax, and up-regulating the anti-apoptotic factor Bcl2 to reduce apoptosis. In addition, berberine can also increase the viability of C17.2 cells by upregulating the expression of the extracellular signal-related kinase (ERK) and phosphorylated extracellular signal-related kinase (p-ERK), activating the WNT/β-Catenin signaling pathway, and increasing the expression levels of pre-neural factors ASCL1, NeuroG1, NeuroD2, and DCX. This further reduces oxidative damage to C17.2 neural stem cells and induces them to differentiate into neurons. Moreover, studies have shown that berberine exerts neuroprotective effects by activating antioxidant enzymes such as superoxide dismutase (SOD) and glutathione (GSH) to antagonize oxidative stress caused by chronic cerebral hypoperfusion [106].

Another alkaloid that can reduce oxidative stress damage in nerve cells is nuciferine. CAT, SOD, and GSH-Px are well-known antioxidant enzymes because they have strong free radical scavenging effects in tissues and cells. Excessive levels of reactive oxygen species (ROS) in cells reduce the activity of these enzymes [107]. However, studies have shown that nuciferine can bring these enzyme activities in diabetic rats induced by alloxan to close-to-normal levels, thereby protecting nerve cells from oxidative damage [108]. It is a potential drug for the treatment of Alzheimer’s disease caused by diabetes by exerting neuroprotective effects. Similar to nuciferine, tetrahydropalmatine can also play an antioxidant role by increasing the levels of SOD, GSH, and CAT and reducing the level of MDA, effectively reducing the oxidative stress injury to nerve cells [109].

### 3.3. Neuroprotection towards Regulating Autophagy

Autophagy is a cellular process of massive degradation of proteins and organelles in cells, which can lead to non-apoptotic programmed cell death called autophagic cell death [110]. The traditional theory holds that after neuronal damage leads to apoptosis, autophagy mediated by lysosomes, as the main organelles, will be induced and enhanced [111]. However, new studies have shown that although autophagy is associated with various cellular damage mechanisms, much apoptosis caused by enhanced autophagy will greatly weaken the autophagy function, leading to the aggravation of damage. This is a dynamic process in neuronal damage [112]. Therefore, the regulation of autophagy function is essential for the protection of neurons. As shown in Figure 4, in a mouse model of traumatic brain injury (TBI), the levels of malondialdehyde (MDA), glutathione (GSH), and glutathione peroxidase 4 (GPX4) in brain tissue were detected by enzyme-linked immunosorbent assay (ELISA). The levels of Beclin 1, light chain 3 (LC3) II/I, p62, GPX4, and ferritin heavy chain 1 (FTH1) were detected by Western blotting (WB) and immunofluorescence. It has been found that tetrandrine could reduce MDA content and increase the GSH content during the period after TBI. It can also reverse the changes in the expression levels of the above autophagy-related proteins after TBI and promote autophagy. Through antioxidative damage and regulation of autophagy, it effectively exerts neuroprotective effects and improves neurological function and reduces brain edema after TBI in mice [113].

In addition, other studies have shown that tetrahydropalmatine can also play a role in autophagy regulation. Studies have found that the expression of Beclin-1 and LC3II/I increased after I/R injury in rats, while the expression of p62 decreased, which confirmed that autophagy was activated after I/R injury. Further studies have found that this is related to the inhibition of the PI3K/AKT/mTOR pathway. Tetrahydropalmatine can reactivate this pathway and reduce the level of autophagy [114]. Nuciferine also has an autophagy regulatory effect. Through the study of the rat pMCAO model, it was found that the levels of autophagy markers decreased, and the accumulation of autophagy substrates was reduced in the early stages of cerebral ischemia. This is related to the inhibition of the autophagy–lysosomal (ALP) pathway mediated by transcription factor EB (TFEB) [115]. In addition, other studies have shown that nuciferine can increase the expression of TFEB and activate the ALP pathway, making the above autophagy markers and autophagy substrates approach normal levels [116]. Although this study only showed that nuciferine regulates autophagy, we believe that regulating autophagy is also an important basis for nuciferine to exert neuroprotective effects.

### 3.4. Neuroprotection towards Calcium Overload

As an essential signaling molecule and cell function regulator, Ca^2+^ has made substantial progress in the study of function and mechanism [117]. It is also accepted that an imbalance in Ca^2+^ homeostasis is closely linked to the development of various human pathologies, including neurodegenerative diseases. Calcium homeostasis is closely related to neurodegeneration, neurotoxicity, neuroinflammation, autophagy, and mitochondrial function changes [118]. As shown in Figure 5, studies have confirmed that methamphetamine (MA) can affect the mitochondrial calcium ATPase responsible for pumping Ca^2+^ into the internal space of mitochondria for storage, directly produce neurotoxicity to cortical cells, and induce cell death with an increase in calcium load. A low dose of morphine can reduce calcium overload induced by MA in PC12 and U87 cells, and significantly improves cell viability by reducing cytotoxicity, caspase-3 activity, and intracellular calcium concentration. The neuroprotective effect of low-dose morphine may also be related to the reduction of inflammatory injury after calcium overload [119]. Berberine also inhibits intracellular calcium overload. Excitatory amino acid toxicity damage leads to the destruction of intracellular calcium balance. Calcium overload is a trigger for oligodendrocyte death. In vitro, the OGD/R ischemia model found that berberine can prevent intracellular calcium accumulation in a concentration-dependent manner, protecting OLN-93 oligodendrocytes from excitotoxicity-induced cell damage [120].

### 3.5. Neuroprotection towards Mitochondrial Dysfunction

Energy metabolism is the foundation of cellular life activities. Mitochondria, known as the engine of human energy conversion, play a major role in cell life activities [121]. A variety of evidence has shown that energy metabolism disorders and neuronal damage caused by mitochondrial dysfunction are the pathological basis of various degenerative neurological diseases, including cerebral ischemia, Alzheimer’s disease, and Parkinson’s disease [122]. Weiyi Li et al. used several cell ischemic injury apoptosis models in their study, including a serum deprivation cell model, a glutamate-induced RGC-5 cell death model, and a hydrogen peroxide (H_2_O_2_)-induced RGC-5 cell death model, as well as an astrocyte-derived neuron-like RGC-5 cell model. It was found that tetrandrine treatment protected staurosporine-induced RGC-5 cells from serum deprivation-induced cell death, significantly increased the relative number of cells cultured with 1 mM H_2_O_2_, and significantly prevented 25 mM glutamic acid-induced cell death in a dose-dependent manner. It has a protective effect on a variety of cells. Further studies have shown that the protective effect of tetrandrine is related to the reduction of mitochondrial transmembrane potential (δψm), improvement of mitochondrial function, inhibition of caspase-3 activation induced by ischemia/reperfusion injury, and reduction of bcl-2 expression [123]. Morphine can also play a similar role in mitochondrial protection. The decrease in mitochondrial membrane potential means that the permeability of mitochondria increases after dysfunction, leading to the release of caspase and nuclease-activating protein. This is a major cause of apoptosis. Studies have shown that morphine can alleviate nicotine-induced mitochondrial dysfunction in PC12 cells and reduce caspase-3 release. As shown in Figure 5, this neuroprotective effect is also associated with a reduction in intracellular calcium levels [124]. There is a complex interaction between calcium overload, mitochondrial damage, and apoptosis [125].

In addition, studies on the zebrafish model of Parkinson’s disease, showed that berberine could easily penetrate the blood–brain barrier. Subcellular localization studies have shown that berberine rapidly and specifically accumulates in the mitochondria of PC12 cells, inhibited the accumulation of Pink1 protein and inhibited the overexpression of LC3 protein in 6-OHDA-injured cells. It is confirmed that mitochondria are the potential sites of berberine in the brain, and berberine may improve the nerve injury caused by Parkinson’s disease by regulating mitochondrial function [126].

### 3.6. Neuroprotective Effects of Promoting Vascular Endothelial Proliferation and Neuronal Regeneration

More and more studies have shown that isoquinoline alkaloids play a significant role in promoting the proliferation of vascular endothelial cells and the regeneration of neurons. By inducing the angiogenesis of endothelial cells and accelerating the recovery of damaged neuronal cell structure and function, it can rapidly improve memory and cognitive dysfunction caused by neuronal damage and improve the prognosis of ischemic stroke [127]. This is still of great significance for the treatment of neurodegenerative diseases by promoting the repair and regeneration of damaged neurons. Based on the clinical experience of traditional Chinese medicine, molecular docking was performed on tetrahydropalmatine and VEGFR2 to demonstrate their binding potential. Metabolomics analysis showed that it can increase the expression of VEGFR2, which is a trigger for angiogenesis and has the potential to promote angiogenesis and exert neuroprotective effects [128]. Studies have also shown that tetrandrine has a role in promoting angiogenesis. Vascular endothelial growth factor A (VEGF-A), which belongs to the same family as VEGFR2, plays an important role in angiogenesis. It has been observed that tetrandrine can increase the expression of VEGF-A mRNA in H9C2 cells. Tetrandrine treatment can increase the synthesis of new VEGF-A mRNA but has no impact on the stability of VEGF-A mRNA. This can enhance the angiogenesis activity of endothelial cells and improve blood flow recovery and capillary density after ischemic limb injury. This is associated with increased VEGF-A expression [129]. It is not only limited to the recovery of ischemic diseases but also of great significance for promoting neuronal repair and regeneration.

## 4. Summary and Outlook

Finally, we comprehensively analyzed the neuroprotective mechanisms of isoquinoline alkaloids and the key targets of their regulation. As shown in Figure 6, we found that the same alkaloid component can exert direct or potential neuroprotective effects in many different ways. These mechanisms are closely related to each other, such as calcium overload, inflammation, and autophagy. There are both cascades and reciprocal causation. This indicates that isoquinoline alkaloids can exert neuroprotective effects through multiple links, multiple targets, and multiple pathways. Some existing studies have also confirmed our idea [130]. We further found that because these isoquinoline alkaloids have a common structural basis, they often reduce neuronal apoptosis through very similar mechanisms of action, with similar or even the same targets. For example, berberine, nuciferine, and tetrahydropalmatine, mentioned above, can increase SOD, GSH, and CAT in vivo to reduce oxidative damage. Tetrandrine and morphine can reduce the release of the apoptotic factor caspase-3. Both tetrahydropalmatine and tetrandrine can act on the same family of vascular endothelial growth factors (VEGF) to promote angiogenesis and neuronal regeneration. In the further analysis of the structural characteristics and pharmacological activities of these alkaloids, we found that both the five isoquinoline alkaloids we discussed in detail, and the dozens of isoquinoline alkaloids we listed in Table 1, have the following feature: the isoquinoline nucleus itself contains a benzene ring, most of these drugs are connected to multiple ether bonds on the benzene ring (most of them have three to four), and these ether bonds have the potential to form hydroxyl groups. These structures themselves directly contain a small number of hydroxyl groups, which means that their polarity is relatively low, which is conducive to their passage through the blood–brain barrier. If they pass through the blood–brain barrier, multiple ether bonds are converted into hydroxyl groups connected to the benzene ring, then these aromatic hydroxyl groups are very effective antioxidant structures. Based on this, we can even speculate that isoquinoline alkaloids may play a neuroprotective role mainly by exerting antioxidant effects combined with anti-inflammation, autophagy regulation, and inhibition of calcium overload. Of course, this conjecture has serious limitations, and more experimental verification is needed in order to provide a sufficient scientific foundation.

The neuroprotective isoquinoline alkaloids and their neuroprotective mechanisms are not limited to those mentioned above. These active effects are the fundamental pathological mechanisms shared by a variety of neurodegenerative diseases, indicating that isoquinoline alkaloids may not only be effective for a single neurodegenerative disease but also have therapeutic effects on a variety of neurodegenerative diseases. At the same time, we also found that they have strong similarities in the mechanism of action, signaling pathways, and target sites, and they share the same structural nucleus. Therefore, we have the following interesting thoughts: (1) The very similar mechanisms and targets of these alkaloids may be the basis for their combined use through different mechanisms to exert neuroprotective effects and synergistic effects in the treatment of neurodegenerative diseases. (2) The way these alkaloids exert neuroprotective effects is often multitarget and multi-channel. Whether or not their properties are better than the current clinical use of single-effect neuroprotective drugs, they may have greater potential for development as neuroprotective drugs. (3) They have the same isoquinoline core structure, which may help to study their structure–activity relationship. Perhaps we can extend the investigation to other components or derivatives with similar structures by studying the structure–activity relationship of one or several drugs.

However, a limiting factor for further research on isoquinoline alkaloids is the lack of basic research on the active components of these alkaloids. Although research on these alkaloids has gradually entered the field of vision of researchers, due to the many and complex mechanisms of these components, basic research on them still lacks systematic and comprehensive study. This undoubtedly brings difficulties for further research and clinical studies on isoquinoline alkaloids. Therefore, we believe that the study of these compounds should pay attention to their similarities and structural features. At the same time, we also believe that strengthening the study of these compounds is helpful in applying the advantages of multitarget and multi-pathway neuroprotective effects to the treatment of neurodegenerative diseases as soon as possible.

Finally, we also hope that this review will enable more researchers to focus on the neuroprotective effects of isoquinoline alkaloids and provide some ideas for conducting more in-depth research in this field.

## Figures and Tables

**Figure 1 molecules-28-04797-f001:**
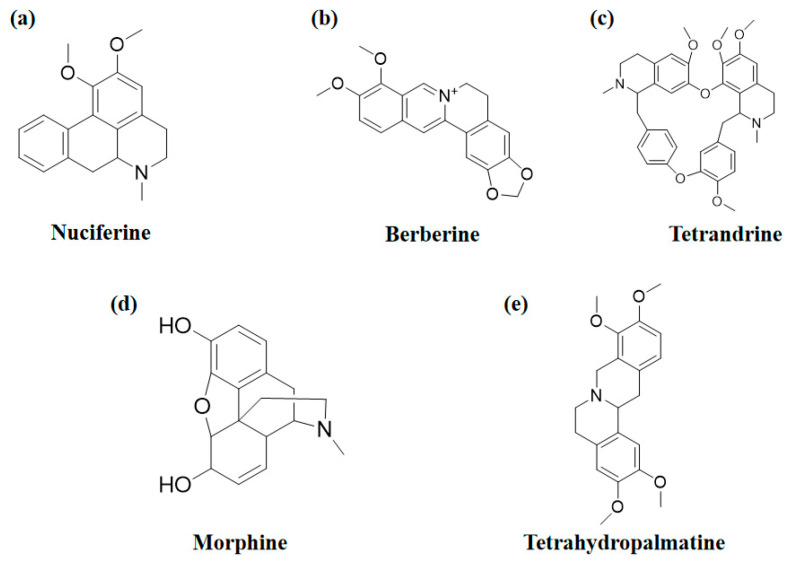
The chemical structure of isoquinoline alkaloids with neuroprotective effects. (**a**–**e**) These are the chemical structures of nuciferine, berberine, tetrandrine, morphine, and tetrahydropalmatine, respectively. They all have the same isoquinoline (benzopyridine) core structure.

**Figure 2 molecules-28-04797-f002:**
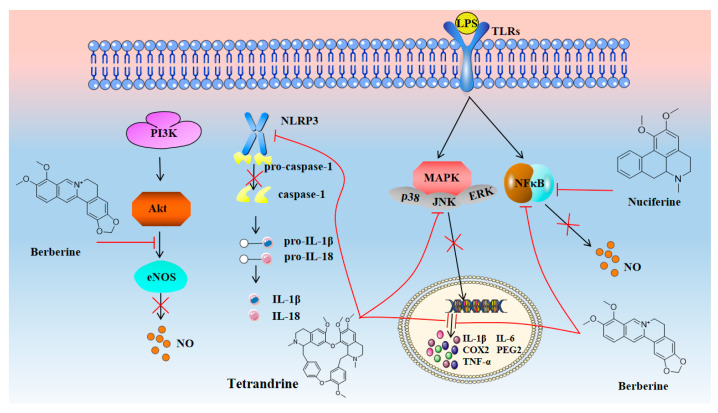
Isoquinoline alkaloids exert neuroprotective effects by reducing inflammatory injury.

**Figure 3 molecules-28-04797-f003:**
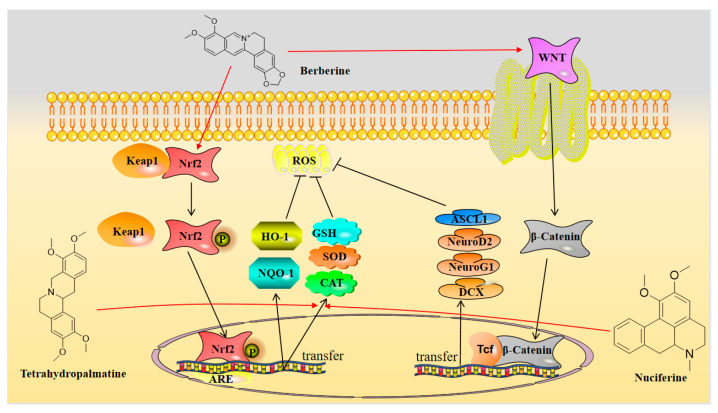
Isoquinoline alkaloids exert neuroprotective effects by reducing oxidative stress.

**Figure 4 molecules-28-04797-f004:**
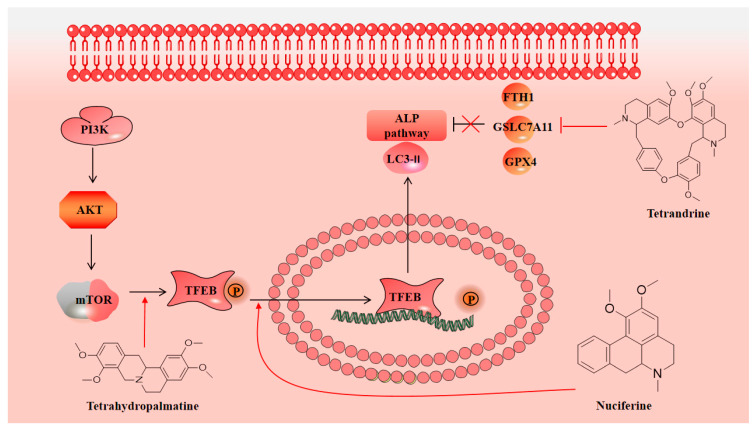
Isoquinoline alkaloids exert neuroprotective effects by regulating autophagy.

**Figure 5 molecules-28-04797-f005:**
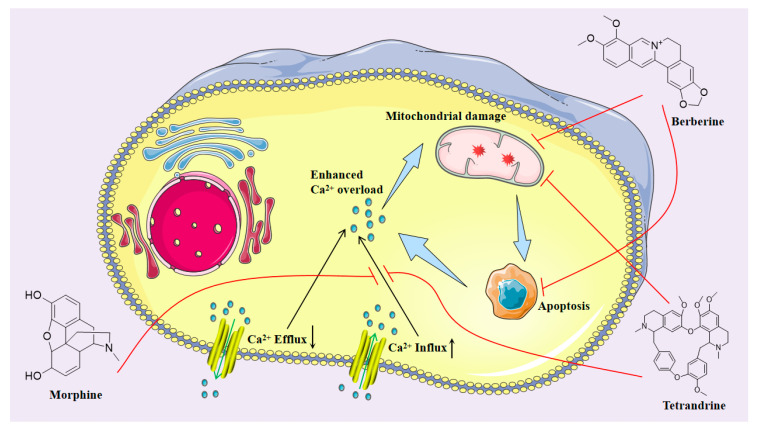
Isoquinoline alkaloids exert neuroprotective effects by alleviating inflammation and mitochondrial dysfunction after calcium overload.

**Figure 6 molecules-28-04797-f006:**
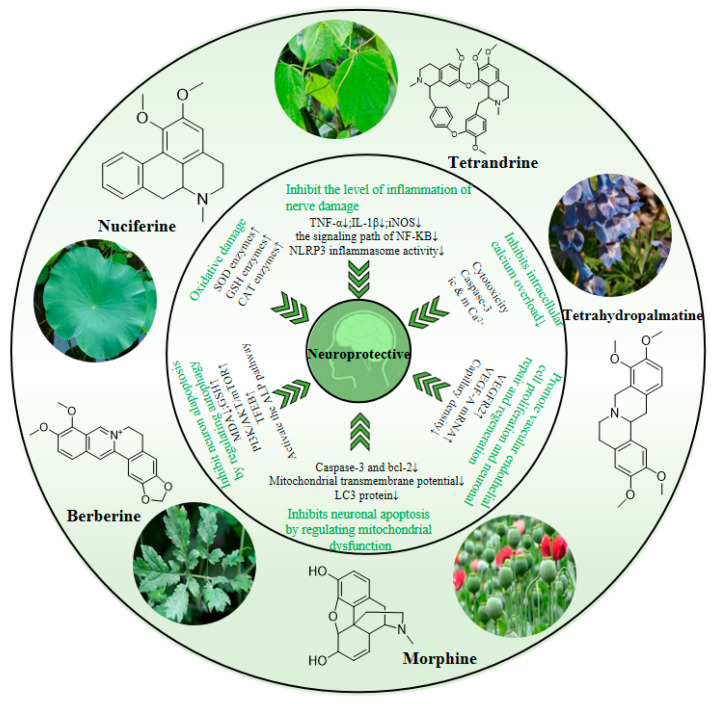
Neuroprotective mechanism of isoquinoline alkaloids.

**Table 1 molecules-28-04797-t001:** Isoquinoline alkaloids with neuroprotective effects and their neuroprotective mechanisms.

Alkaloid	Structure	Neuroprotective Mechanism	Reference
Papaverine	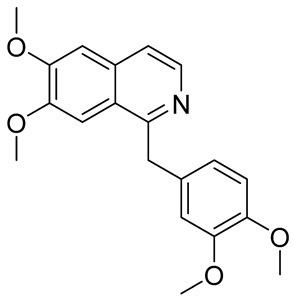	anti-inflammatory; anti-oxidation; anti-apoptosis; promote neurogenesis; inhibition of α-synuclein aggregation	[41,42,43,44,45,46]
Higenamine	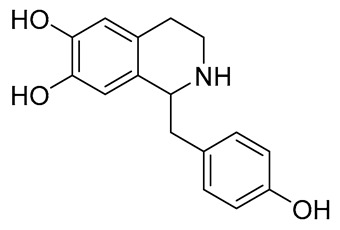	anti-inflammatory; anti-oxidation; anti-apoptosis;	[47,48,49]
Sinomenine	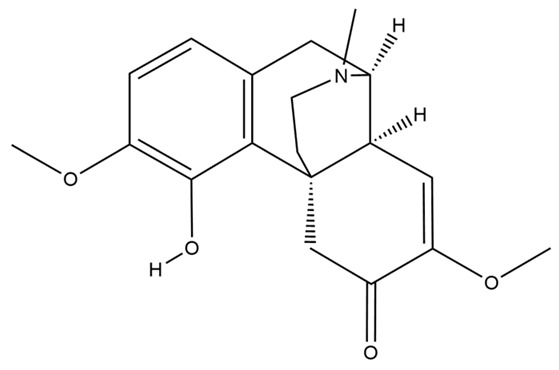	anti-inflammatory; anti-oxidation; regulating autophagy; anti-pyroptosis; anti-apoptosis; neuroimmune intervention; inhibition of Ca^2+^ overload	[50,51,52,53,54,55,56,57,58]
Sanguinarine	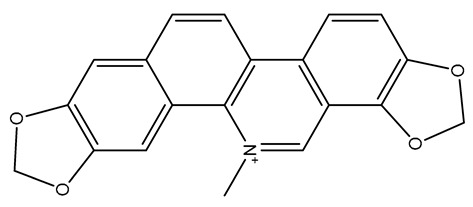	anti-inflammatory; anti-apoptosis; mitochondrial protection; inhibition of Ca^2+^ overload	[59,60,61,62]
Neferine	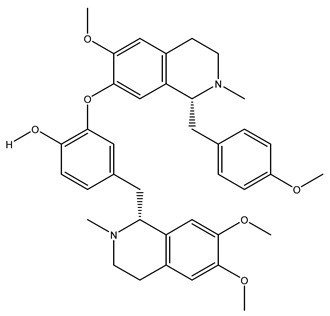	anti-inflammatory; anti-oxidation; anti-apoptosis; regulating autophagy; inhibition of Ca^2+^ overload; mitochondrial protection	[63,64,65,66,67,68,69,70]
Stepharine	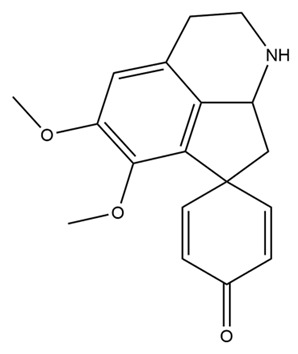	anti-inflammatory; anti-apoptosis; anti-oxidation	[71,72,73]
Dauricine	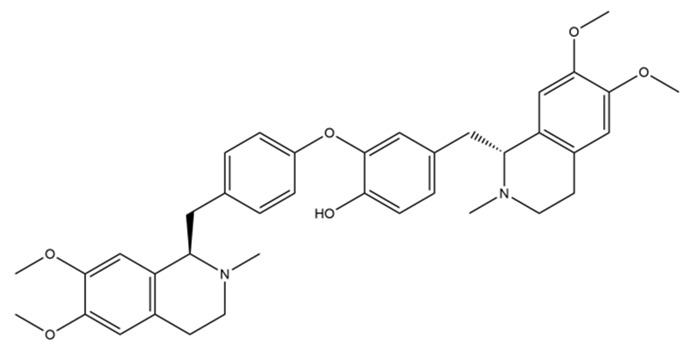	anti-inflammatory; anti-oxidation; anti-apoptosis; acceleration of Aβ protein degradation; inhibition of ferroptosis; enhance mitochondrial function	[74,75,76,77,78]
Lycorine	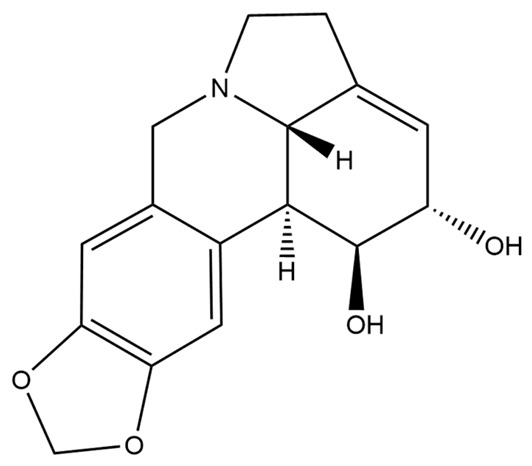	anti-oxidation; anti-apoptosis;	[79,80]
Piperine	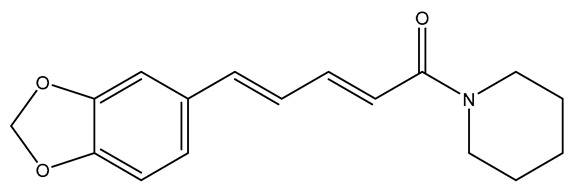	anti-inflammatory; anti-oxidation; anti-apoptosis; improve mitochondrial dysfunction; reduce the toxicity of excitatory amino acids; up-regulate nerve growth factor	[81,82,83,84,85,86,87]
Jatrorrhizine	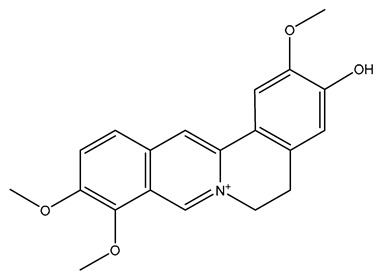	anti-inflammatory; anti-oxidation; anti-apoptosis; improve vascular endothelial dysfunction	[88,89,90]

## Data Availability

Not applicable.

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
