# Peer review of "Research Progress on Neuroprotective Effects of Isoquinoline Alkaloids"

_molecules, 2023, doi:10.3390/molecules28124797_

Round 1
Reviewer 1 Report
Kindly see the attachment.

Minor editing is required.
Reviewer 2 Report
In this paper, the authors summarize some active components of isoquinoline alkaloids with neuroprotective effects, expound the different mechanisms of their neuroprotective effects, and summarize their characteristics to provide a reference for further study on the neuroprotective effects of these isoquinoline alkaloids. It involves 123 articles, with rich research content and large workload, which can provide readers with the latest progress in the field. Therefore, the article can be accepted after minor revision.
In introduction: The review literature retrieval media, key words, etc. should be given. And clarify whether there are similar review articles before and the relationship with this article.
The article only describes the research progress of alkaloids, which has certain limitations. It is recommended to supplement the research progress of related alkaloid derivatives, especially some derivatives that have entered clinical research or have been commercialized, which can increase readers' reading interest.
The structural formulas of these compounds are somewhat non-standard. It is recommended to standardize the structures of these compounds according to the requirements of the journal.
There are grammar and spelling errors in some places that need to be carefully corrected.
There are grammar and spelling errors in some places that need to be carefully corrected.
Reviewer 3 Report
We consider the manuscript pertinent to the readers of this Journal’s Special Issue “Natural Products in Asia” of the Section “Natural Products Chemistry”.
The objectives of this work are briefly specified in the abstract. However, a clear and comprehensive justification of the need for a new review of the theme is lacking.
Some similar reviews published recently were found, although do not cited in the manuscript. The main reviews are itemized below and the fundamental Keywords were used to illustrate the gap of knowledge, to help the reasoning, and support the relevance of a new review on the subject. In our opinion, this manuscript indeed provides a deeper discussion of autophagy and hypercalcemia aspects.
1. This manuscript_ ->”neuroprotective” keyword ->71x; inflammatory keyword ->42x; oxidative stress->14x; autophagy -> 50x; Hypercalcemia(calcium overload) -> 8x;
2. Kong, Y.R.; Tay, K.C.; Su, Y.X.; Wong, C.K.; Tan, W.N.; Khaw, K.Y. Potential of Naturally Derived Alkaloids as Multi-Targeted Therapeutic Agents for Neurodegenerative Diseases. Molecules 2021, 26, 728. https://doi.org/10.3390/molecules26030728 (review) ->”neuroprotective” keyword ->15x; inflammatory keyword ->5x; oxidative stress->12x; Hypercalcemia -> (1x);
3. Erika Plazas, Mónica C. Avila M, Diego R. Muñoz, Luis E. Cuca S. Natural isoquinoline alkaloids: Pharmacological features and multi-target potential for complex diseases. Pharmacological Research 177 (2022) 106126. https://doi.org/10.1016/j.phrs.2022.106126- (review) >”neuroprotective” keyword ->39x; inflammatory keyword ->73x; oxidative stress->11x; autophagy -> 12x
4. Hussain G, Rasul A, Anwar H, Aziz N, Razzaq A, Wei W, Ali M, Li J, Li X. Role of Plant Derived Alkaloids and Their Mechanism in Neurodegenerative Disorders. Int J Biol Sci. 2018 Mar 9;14(3):341-357. doi: 10.7150/ijbs.23247. (review) ->”neuroprotective” keyword ->19x; inflammatory keyword ->23x; oxidative stress->15x; autophagy -> 10x
5. Cahlíková L, Vrabec R, Pidaný F, PeÅ™inová R, Maafi N, Mamun AA, Ritomská A, Wijaya V, Blunden G. Recent Progress on Biological Activity of Amaryllidaceae and Further Isoquinoline Alkaloids in Connection with Alzheimer's Disease. Molecules. 2021 Aug 29;26(17):5240. doi: 10.3390/molecules26175240. (review) ->”neuroprotective” keyword ->25x; inflammatory keyword ->3x; oxidative stress->4x;
6. Xiao-Fei Shang, Cheng-Jie Yang, Susan L. Morris-Natschke, Jun-Cai Li, Xiao-Dan Yin, Ying-Qian Liu, Xiao Guo, Jing-Wen Peng, Masuo Goto, Ji-Yu Zhang, Kuo-Hsiung Lee. Biologically active isoquinoline alkaloids covering 2014–2018. Medicinal Research Reviews. 2020. Volume40, Issue6. Pages 2212-2289 https://doi.org/10.1002/med.21703 (review)->”neuroprotective” keyword ->5x; inflammatory keyword ->36x; oxidative stress->9x; autophagy -> 5x
# Question 1: Dear authors, is there an explanation for the omission of such relevant reviews?
The introduction, although quite brief, covers the core items. The statements and conclusions drawn throughout the manuscript, are coherent and supported by the listed citations, and perfectly integrate the theme's main aspects.
We found this article well written, with a good organization of the contents. Some very nice and creative graphic pics, with good quality, accompanying the discussion increased the understanding of the discussed theme and clarified the reasoning. We congratulate the authors on that!
